# In Vitro Evaluation of the Potential Anticancer Properties of Cu-Based Shape Memory Alloys

**DOI:** 10.3390/ma16072851

**Published:** 2023-04-03

**Authors:** Minja Miličić Lazić, Marko Lazić, Jelena Milašin, Danica Popović, Peter Majerič, Rebeka Rudolf

**Affiliations:** 1School of Dental Medicine, University of Belgrade, 11000 Belgrade, Serbia; minja.milicic@stomf.bg.ac.rs (M.M.L.); marko.lazic@stomf.bg.ac.rs (M.L.); jelena.milasin@stomf.bg.ac.rs (J.M.); danica.popovic@stomf.bg.ac.rs (D.P.); 2Faculty of Mechanical Engineering, University of Maribor, 2000 Maribor, Slovenia; peter.majeric@um.si

**Keywords:** Cu-Al-Ni SMA, microstructure, phases, properties, osteosarcoma cells, anticancer properties, cells morphology

## Abstract

Due to the unique functional properties of shape memory alloys (SMAs) and current scientific interest in Cu-containing biomaterials, a continuously cast Cu-Al-Ni alloy in the form of rods has been investigated as a potential candidate for biomedical application. Additionally, the fact that Cu- complexes have an antitumour effect served as a cornerstone to develop more efficient drugs based on trace element complexes. In line with that, our study aimed to analyse the basic properties of the Cu-Al-Ni alloy, along with its anticancer properties. The detailed chemical analysis of the Cu-Al-Ni alloy was performed using XRF and SEM/EDX analyses. Furthermore, a microstructural and structure investigation was carried out, combined with hardness measurements using the static Vickers method. Observations have shown that the Cu-Al-Ni microstructure is homogeneous, with the presence of typical martensitic laths. XRD analysis confirmed the presence of two phases, β′ (monoclinic) and γ′ (orthorhombic). The viability of osteosarcoma cells in contact with the Cu-Al-Ni alloy was evaluated using epifluorescence microscopy, while their morphology and attachment pattern were observed and analysed using a high-resolution SEM microscope. Biocompatibility testing showed that the Cu-Al-Ni alloy exerted a considerable antineoplastic effect.

## 1. Introduction

Shape memory alloys (SMAs), also called intelligent functional alloys, are materials known for their distinctive feature of adapting their shape to the conditions of the external environment. This unique property, different from ordinary metallic materials, is based on the reversibility of the crystallographic lattice analogous to steel technology [1,2,3]. Specifically, the temperature-dependent phases of these metallic materials are austenite and martensite. The crystallographic transformations (austenite ⇄ martensite) are triggered by temperature change or external force. Depending on the external stimuli, two potentially functional properties can appear, shape memory effect and superelasticity. If the material is subjected to an appropriate thermal procedure, then it returns to the previous shape thanks to the shape memory effect (SME). If alloys exhibit SME only upon heating it is called one-way shape memory, whereas if the material shows SME during both heating and cooling, it possesses two-way shape memory [4]. On the other hand, superelasticity is the phenomenon of recovering the original shape after removing the mechanical force that led to the deformation. Anyway, in both cases these macroscopic changes are characterised by the coordinated movement of groups of atoms, called martensitic transformation.

Due to martensitic transformation, SMAs have found a wide range of applications, from aerospace industries to medical devices. To date, different types of shape memory alloys have been discovered: Ni-Ti, Ni-Ti-Hf, Ni-Mn-Ga, Cu-Al-Ni, Cu-Zn-Al, Au-Cd, Fe-Mn-Si, etc. At the moment, only Ni-Ti and Cu-Al-Ni are available for commercial exploitation. Owing to the great scientific interest in Cu-containing biomaterials, the Cu-Al-Ni alloy is being investigated as a potential replacement for the widely known Ni-Ti shape memory alloy.

The literature is consistent in the conclusion that the biofunctional properties of Cu-based alloys need to be examined comprehensively in the context of the materials production processes [5,6,7]. In recent years, several very successful studies have been accomplished with continuous casting (small dimension rods of Au-and Ag–based alloys) [8,9,10]. Continuous casting has also been studied in the frame of Ni-Ti alloy, but numerous technical limitations have been encountered, while, on the other hand, Cu-based alloys have the advantage of being less difficult to produce by casting. Metal-based antitumour drugs have been used as adjuvant therapy since the 1980s. The most common are cisplatin (cis-diamine) and platinum [11]. Due to their side effects, toxicity and resistance phenomena, contemporary research has focused on finding biologically effective complexes based on trace elements, such as Cu [12]. Cu-complexes differ widely in their effect on tumour cells in comparison to Cu ions. Specifically, due to the proven angiogenic potential of Cu ions, high doses of Cu ions favour tumour growth. Numerous studies have shown elevated Cu values in tumour tissue as evidence for this claim [13,14]. Contrarily, the molecular mechanisms of Cu-complexes are based on covalent binding to the DNA of cancer cells. When adducted covalently to the DNA helix, these Cu complexes exert an inhibitory effect on topoisomerases, the enzymes responsible for DNA replication, which leads to oxidative damage, known as a “stress condition” for cancer cells [11].

Based on the presented problem of different biomaterials where the main component in the material is Cu, the idea of this research was to test a continuously cast Cu-Al-Ni rod in the form of discs for potential use in very demanding special biomedical cases, where, due to the presence of Cu-complexes, they could give rise to the opposite effect, i.e., display antineoplastic properties, which has not been exploited until now.

Hence, the present study aimed to investigate several physicochemical and mechanical properties of the material, along with its biocompatibility. Specifically, we examined the chemical composition and surface microstructure of the Cu-Al-Ni alloy, and determined its hardness and microhardness. Additionally, we also assessed the potential cytotoxic effects of this biomaterial on human cancer cells (an osteosarcoma cell line).

## 2. Materials and Methods

### 2.1. Samples Preparation

The material used in the present study was a continuously cast Cu-Al-Ni rod with the diameter 2r = 9 mm [15]. The Cu-Al-Ni alloy was prepared by vacuum induction melting of pure Cu, Al and Ni components immediately before casting. Experimental casting at about 1300 °C was performed with a laboratory scale vertical continuous casting (VCC) device, TechnicaGuss, which was connected to the 60 kW medium-frequency (4 kHz) vacuum induction melting (VIM) furnace LeyboldHereaus. The charge was approximately 15 kg. The induction power was 10 kW for the first 10 min; for the next 10 min 20 kW; and, in the final 5 min, 30 kW. Melting was carried out in a vacuum. Just before the start of casting, the chamber was filled with argon (purity 5.0). To minimise the risk of sticking and fracture of the rod in the mould, relatively low casting rates were selected, and the pulling sequence was programmed with the lowest possible acceleration. The cast rod was cut by electrical discharge machining (EDM) into discs (thickness 1.7 mm) suitable for further in vitro experiments.

The metallographic preparation followed the embedding of the discs in a hot-mounting mass, which made it possible to grind and polish the upper surface of the disc sample. The grinding process was carried out with abrasive SiC papers in grades of 180–4000 on the grinding/polishing machines BUEHLER Automet 250 and EcoMet 250, (Lake Bluff, IL, USA) while polishing was done with C paste sizes 1 and 0.5 µm. Finally, the samples were cleaned with acetone, alcohol and deionised water. For the microstructure observations all samples were etched with a chemical reagent that had the composition 5 g FeCl_3_, 10 mL HNO_3_, 100 mL H_2_O–, while the etching time was 30 s.

### 2.2. Determination of the Chemical Composition and Microstructure/Phase Characterisation

X-ray fluorescence (XRF) spectrometry was used to analyse the chemical composition. A Niton XL3t GOLDD+ XRF Analyzer (USA),equipped with an Ag anode of 50 kV and a 200 μA tube type as a source of X-rays, and a Geometrically Optimised Large Area Drift Detector were used for the measurement.

The semi-quantitative microchemical analysis was done by high resolution Scanning Electron Microscopy (SEM), Sirion 400NC (FEI, Hillsboro, OR, USA), with an Energy Dispersive X-ray spectroscope (EDX) INCA 350 (Oxford Instruments, Oxford, UK). For this purpose, the analysis was performed directly on the surface of the disc samples without metallographic preparation, in order to exclude the possible influence of the etching reagent on the results. EDX chemical analyses were performed on the central part of the samples, by selecting two measuring segments, and, within each of them, mapping four measuring points. In the marginal part of the sample, measuring was performed at four points of the measuring segment.

The microstructure of the Cu-Al-Ni discs was investigated with an optical metallographic microscope, NIKON Epiphot 300 (Japan), with an Olympus DP12 camera (Boston, MA, USA). A Scanning Electron Microscope (SEM), Sirion 400NC (FEI, Hillsboro, OR, USA), with an Energy-Dispersive X-ray (EDX) spectroscope, INCA 350 (Oxford Instruments, Oxford, UK), was used for the detailed microstructure observation.

The additional analysis of the phases was performed with X-ray diffraction (XRD) on Cu-Al-Ni discs with a Panalytical XPERT Pro PW 3040/60 goniometer (Panalytical, Almelo, TheNetherlands), 2theta 10–90° with a step of 0.002° and a time of 100 ms per step. The anode was Cu (Kalfa = 0.154 nm) with a current of 40 mA and a voltage of 45 kV.

### 2.3. Hardness Measurements

The hardness and microhardness were measured using the static Vickers method, according to the ISO/R 399:1964 Standard for Cu-based alloys [16]. The measurements were performed on the polished surfaces of the discs, which were prepared as described in part 2.1. The testing procedures were as follows:(a)For the hardness measurement, a WPN HPO 250 machine (Leipzig, Germany)was used, which applied the nominal value of a 49.03 N test force load for 5 s.(b)For the microhardness measurement, a Zwick-Roell ZHV10 hardness tester (Kennesaw, GA, USA) was used, and the force applied was 0.49 N for 5 s indentation time.

### 2.4. Biocompatibility

#### 2.4.1. Cell Culture (MG-63 Osteosarcoma Cells)

Human osteosarcoma cells (MG-63, ATCC^®^ CRL-1427™) were grown in a complete growth medium (Dulbecco’s modified Eagle’s medium (DMEM) with 4 mM L-glutamine, 10% Foetal bovine serum (FBS), and 1% ABAM, Sigma-Aldrich, Steinheim, Germany).

#### 2.4.2. Cell Adhesion and Viability of MG-63 Cells

The Cu-Al-Ni discs (*n* = 6), were compared with control samples of round cover glass with the same dimeter as the test samples (*n* = 6, 2r = 9 mm). All of the samples were sterilised with UV-C light for 1 h and transferred to a culture flask with 12 wells.

The MG-63 cells (2 × 10^4^ cells/cm^2^) were seeded in 12 well plates (in direct contact with the samples), and incubated at 37 °C in a humidified 5% CO_2_ atmosphere. After 24 h (when the cells were attached to the samples and had begun to divide) and 7 days of incubation, the samples were prepared for examination with an epifluorescence microscope.

The number of MG-63 cells that adhered to the material surface and their viability were determined using differential staining. The samples with attached cells were washed with Phosphate-Buffered Saline (PBS) to remove floating and unattached cells and stained with 2 μg/mL Hoechst 33342 dye for 20 min. Hoechst 33342 stains the nuclei of live cells with blue fluorescence. The samples were examined and imaged using an epifluorescence microscope (Axio Imager Z1, Carl Zeiss, Germany). The total number of viable cells (blue fluorescence) was counted using ImageJ (NIH). For each type of sample, three plates were examined and cells attached to at least 2.5 mm^2^ were counted for each plate.

#### 2.4.3. Preparation of the Samples for SEM Observation

The morphology of the MG-63 cells attached to the samples was examined using an SEM microscope JEOL JSM-6500F (Tokyo, Japan). Samples with attached cells were fixed, contrasted and dehydrated according to the previously described methodology [17]. Samples were gold coated using a Precision Etching Coating System (682 PECS, Gatan, Pleasanton, CA, USA). The cells were examined on the control (cover glass) and Cu-Al-Ni discs using the morphology and attachment pattern of the adherent MG-63 cells.

### 2.5. Statistical Analysis

The data were analysed using the IBM SPSS Statistics v22 software (SPSS Inc., Chicago, IL, USA). The results of the samples hemical composition were shown as quantitative data and as Mean ± Standard Deviation (SD). The Shapiro-Wilk test was used to determine if the data were normally distributed for the purpose of comparing the results of cell viability. An independent-samples *t*-test was used to evaluate the differences of cells proliferation between the Cu-Al-Ni discs and the control samples. A *p*-value of less than 0.05 was considered to be statistically significant.

## 3. Results

### 3.1. Chemical Composition and Microstructure

The XRF analysis (Table 1) revealed a similar chemical composition to the point EDX analyses, i.e., 12 wt.% Al, about 3.9 wt. Ni and 84 wt.% Cu-, see Table 2. However, therewas also Fe. A small Fe concentration was expected, as it dissolves from the starter tip made from steel during the melting of the metals (Cu, Al, and Ni). Additionally, it is known that Fe is an impurity in some metals, such as Ni. According to the ASTM-F 2063 Standard for medical devices, this production method allows low impurity levels [18].

The characteristic microstructures of the Cu-Al-Ni discs are shown in Figure 1a–c. As can be seen, it is distinctly martensitic, with lamellae around 50 µm long. No inclusions or other defects were observed in the microstructure (such as solidification porosity, other porosity, inclusions, etc.). The martensitic lamellae were oriented. Point chemical analyses were carried out at selected locations, two central measuring segments, and, within each of them, four measuring points (Figure 1a,b), along with the marginal part of the sample (Figure 1c). The results showed that the chemical composition was homogeneous, as the SD value was quite low, and since the presence of other elements was not detected, as shown in more detail in Table 2 by the content for each element.

The optical microstructure of a Cu-Al-Ni disc is shown in Figure 2, along with a simulation of the placement of the grain boundaries. The microstructural ASTM analysis showed that the grain number (G) was 3, and that there were 64 grains per 1 mm^2^, indicating a coarse grain structure.

The results of the ASTM analysis are shown in more detail in Table 3. At G = 3, the estimated grain size was between 62.5–125 µm.

The XRD structural analysis confirmed the presence of two phases (Figure 3): β′ (monoclinic) and γ′ (orthorhombic). These two phases both represent martensite being thermally induced (note: the dominant β′ phase had a lower proportion of Al, i.e., a little smaller than 11.9 wt.%, while the γ′ phase had a slightly higher Al content (>12 wt.%) [19].

### 3.2. Hardness and Microhardness

The results of the hardness measurements, expressed in HV units, are given in Table 4.

### 3.3. Biocompatibility

#### 3.3.1. Fluorescence Microscopy

Figure 4 presents the mean number of viable MG-63 cells per mm^2^. After 24 h differential staining revealed significant differences in the number of viable cells between the control and Cu-Al-Ni discs, emphasising initial surface non-toxicity (Figure 4a). After 7 days of exposure there was a statistically significant decrease of cells viability on the Cu-Al-Ni disc (Figure 4b).

A fluorescent blue stain was used to detect the nuclei of viable cells (Figure 5).

#### 3.3.2. SEM Observations of MG-63 Cells after 24 h

The distribution and the adhesion pattern of cells after 24 h are visible in Figure 6. In both discs, cells with well-visible cellular and membrane outgrowths are predominant, but the morphology of the cells is different between the control and Cu-Al-Ni disc. The cells growing on the Cu-Al-Ni are elongated with several filopodia attached to the disc’s surface, while on the control disc there are more round-shaped cells, with a smaller cell footprint and shorter filopodia, suggesting poorer cell adhesion. Cells with apoptotic and necrotic morphology were not detected at 24 h.

#### 3.3.3. SEM Observations of MG-63 Cells after 7 Days

It is noteworthy that, after seven days of exposure, the cells exhibited substantially different attachment patterns and morphology compared to the initial observation (Figure 7). On the Cu-Al-Ni disc, round-shaped cells were predominant. Large cell extensions were not visible, suggesting poor adhesion. Additionally, many apoptotic bodies could be observed. This was in sharp contrast with the control disc, where the majority of cells were elongated, with numerous cellular interconnections, i.e., with high cell–cell contact.

## 4. Discussion

Several studies have demonstrated that the Cu-Al-Ni alloy can, potentially, be used for the preparation of medical devices [20,21,22], as their results point to better control of chemical composition in comparison to NiTi alloy, higher composition control via melting and casting, ageing resistance, and better stability of the two-way shape memory effect [23].

The toxicity of metal biomaterials is proportional to the released metal ions. Depending on the chemical composition and fabrication, Cu-based alloys can exert different cytocompatibility. Prior to this investigation, we performed a biocompatibility study, and the results indicated that the Cu-Al-Ni alloy did not elicit cytotoxicity against two human cell lines, measured on primary cytotoxicity tests.

In biomedical engineering several attempts have been made to develop a Micro-Electro-Mechanical System (MEMS) using Cu-based shape memory alloys [24,25,26]. It is known that those devices can be constructed from various materials (silicone, polymers, ceramic), but a big step forward was made with the advent of intelligent BioMEMS. The working principle of these bioactuators and sensors is based on the two-way shape memory effect (TWSME). These devices develop micro-mechanical movements, triggered by external stimuli changes (temperature, electromagnetic or fluid-driven). The uniqueness of these stimuli-response devices is their ability to sense and actuate, thanks to the materials’ phase change.

In this study, Cu-Al-Ni continuously cast rods in the form of discs were used for testing. For chemical composition identification we performed the EDX technique, and additional X-ray fluorescence analysis for detailed quantitative chemical information. The results indicated that the obtained chemical composition is close to the optimal one, which is about 13 wt.% Al and 3–4.5 wt.% Ni [23]. According to the literature, if the Ni content increases, the brittleness of the alloy increases, and the eutectoid point shifts to higher Al contents, which has a negative impact on its biofunctional properties [23].

Moreover, alloys with a low percentage of Al and Ni contain a Ƴ2 phase (intermetallic compound (Cu9Al4) precipitations are observed), resulting from the deposition of sediment with an inhomogeneous structure [27].

Taking into consideration the conventional casting manufacturing of the Cu-Al-Ni rods, two possible forms of martensite may be present, depending on the aluminium content.

β′ martensite is predominant in CuAlNi alloys that have more than 13 wt.% aluminium content. Alloys containing less than 13 wt.% aluminium are richer in γ′ martensite [28,29]. Since the Al content in our alloy almost equals the threshold Al content, XRD analysis confirmed the presence of both martensite types. β′_1_ martensite, also known as 18R (rhombohedral) martensite, is characterised by needle-like grain shapes, and it has a very high-thermoelastic behaviour. On the other hand, γ′_1_ martensite, also known as 2H (hexagonal), consists of a plate-like (coarse variants) phase.

Regarding the microstructure properties of the Cu-Al-Ni discs, ASTM analysis confirmed that the continuous casting route of Cu-Al-Ni discs results in semifinished products with a coarse grain structure. This can lead to a decrease in the mechanical properties of the examined alloy, which can affect its potential application negatively. Various studies have shown that coarse-grained alloys have low strain recovery [30,31,32]. The static Vickers test with different loading forces showed that the hardness and microhardness values (Mean = 253 for HV5 and Mean = 280 for HV0.05) were in agreement with earlier published data in studies which investigated the mechanical properties of conventional casting of Cu-Al-Ni alloys [33,34]. In order to get a fine-grained alloy, two methods for microstructure refinement are possible. The most common procedure is adding a fourth alloying element, such as Ti, Zr, Cr, etc. The second method is related to the manufacturing procedures, such as rapid solidification, powder metallurgy and equal channel angular extrusion [35].

Our study also showed that the alloy exerted a clear anticancer effect, which only became apparent after 7 days. Namely, after 24 h, the number of MG-63 cells on the alloy was higher in comparison to cells attached to the control sample. The SEM micrographs showed that elongated cells, oriented along the indentations on the surfaces of the samples, were predominant on the Cu-Al-Ni discs, which suggested normal cell growth and cellular affinity to the tested material. The initial material’s cytocompatibility was proportional to the cells’ ability to attach and spread over its surface; it is called “passive adhesion“. The attachment pattern could also be explained by the fact that osteoblast-like cells possess affinity for surfaces with higher roughness in comparison to smoother ones [36]. Hence, this initial cell’s ability to attach and spread over the material’s surface was more related to its texture than to the chemical composition of the material.

After 7 days of cultures the number of viable cells decreased sharply, and, contrary to cell behaviour after 24 h of culture, the morphological characteristics of the osteosarcoma cells after 7 days pointed to a high level of apoptosis. To be precise, there was a remarkable change of cellular phenotype between 24 h and 7 days of exposure to the alloy. From being elongated, the cells became round-shaped and displayed an increased number of extracellular vesicles, which, given the dimensions (1 to 5 μm), corresponded mostly to apoptotic bodies. The latter are considered to be crucial indicators of cancer cell death [37,38]. The changes recorded after 7 days of cells’ exposure to the alloy might be explained by the chemical composition of the tested material (vs. changes recorded after 24 h, which were due mostly to the surface texture). Certainly, more in-depth studies are needed in order to understand the mechanism of the anti-neoplastic effects of Cu-based alloys.

## 5. Conclusions

The following scientific conclusions can be drawn from the study of Cu-Al-Ni continuously cast discs:The microstructure is homogeneous, with the presence of martensite lamellae 50 µm long, grain size G = 3, and the average grain size was estimated between 62.5–125 µm. This represents a coarse-grained microstructure of the Cu-Al-Ni disc.The chemical composition of the discs, determined by EDX analyses, differed minimally from the chemical composition obtained by XRF analysis, where an Fe content (0.03 wt.) was detected additionally. The content of Fe was negligible, and did not affect the other properties significantly.The XRD structural analysis showed the presence of phases: β′ (monoclinic) and γ′ (orthorhombic).The hardness and microhardness of HV had comparable values and were in the range of 229–290.Cu-Al-Ni discs can achieve an anti-neoplastic effect in the selected environment, which is clearly visible after 7 days on the SEM micrographs. The number of viable cells decreased sharply, and, contrary to cell behaviour after 24 h of culture, the morphological characteristics of the osteosarcoma cells after 7 days pointed to a high level of apoptosis.

However, based on the preliminary results of this study, Cu-Al-Ni alloy could, tentatively, be taken into consideration as a novel pro-apoptotic agent.

## Figures and Tables

**Figure 1 materials-16-02851-f001:**
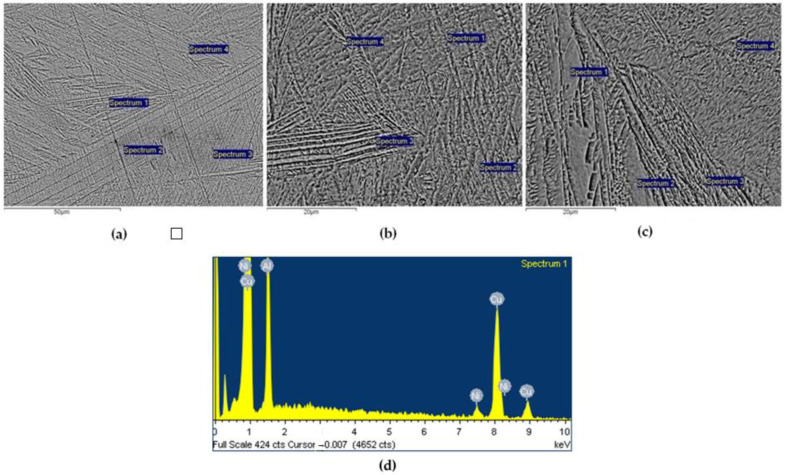
Microstructures and presentation of the points where EDX analyses were performed on the Cu-Al-Ni discs surfaces: (**a**) First measuring segment of the central part of the sample; (**b**) Second measuring segment of the central part of the sample; (**c**) Measuring segment in the marginal part of the sample; and (**d**) Representative EDX graph of Spectrum 1 from the second measuring segment.

**Figure 2 materials-16-02851-f002:**
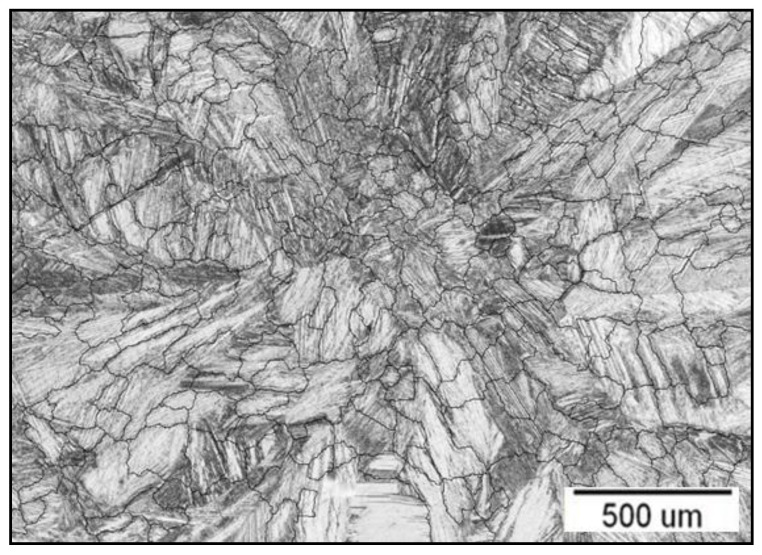
Optical Cu-Al-Ni microstructure with simulation of the grains and their boundaries.

**Figure 3 materials-16-02851-f003:**
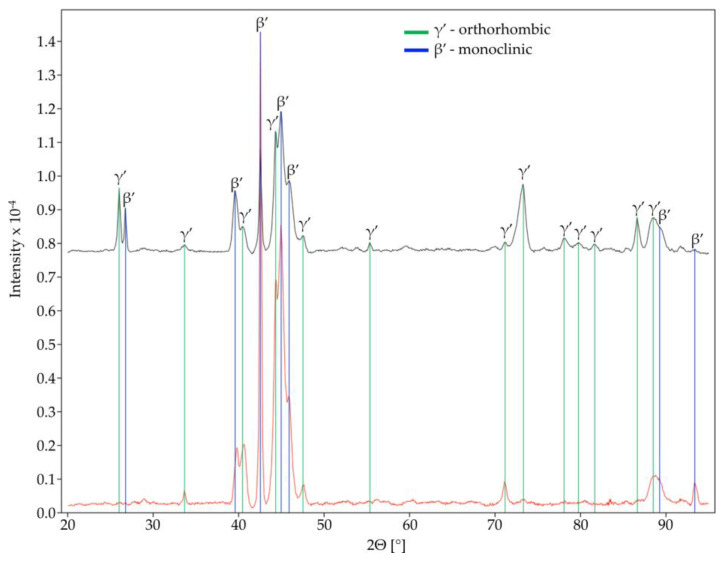
XRD spectra of the Cu-Al-Ni discs.

**Figure 4 materials-16-02851-f004:**
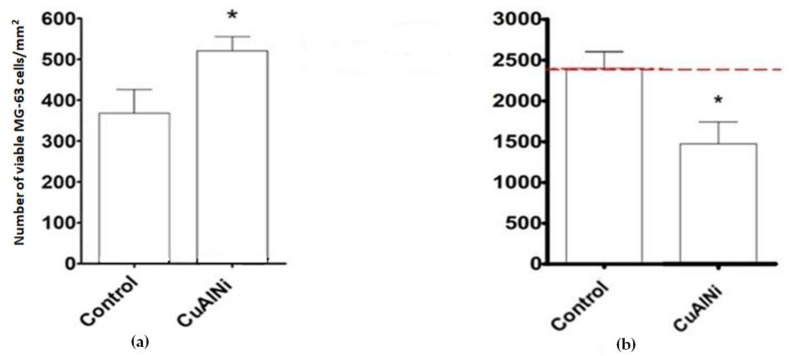
Number of viable cells: (**a**) 24h of exposure. The graph shows the average number of viable MG-63 cells per mm^2^ of disc area (+SD). An asterisk (*) shows a statistically significant difference (*p* < 0.05) compared to the control; (**b**) 7 days of exposure. Viability staining showed a statistically significantly lower number of live cells attached to the Cu-Al-Ni disc.

**Figure 5 materials-16-02851-f005:**
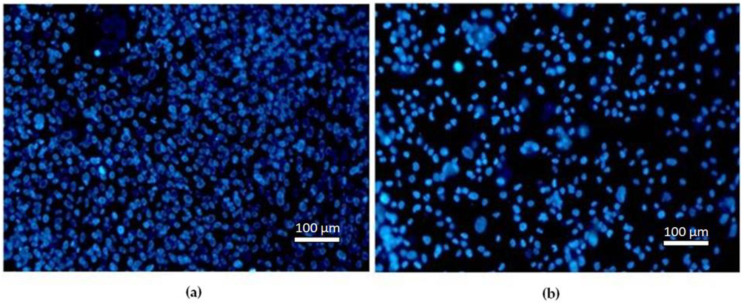
Fluorescent microscopy image: (**a**) Control disc; (**b**) Cu-Al-Ni disc after 7 days.

**Figure 6 materials-16-02851-f006:**
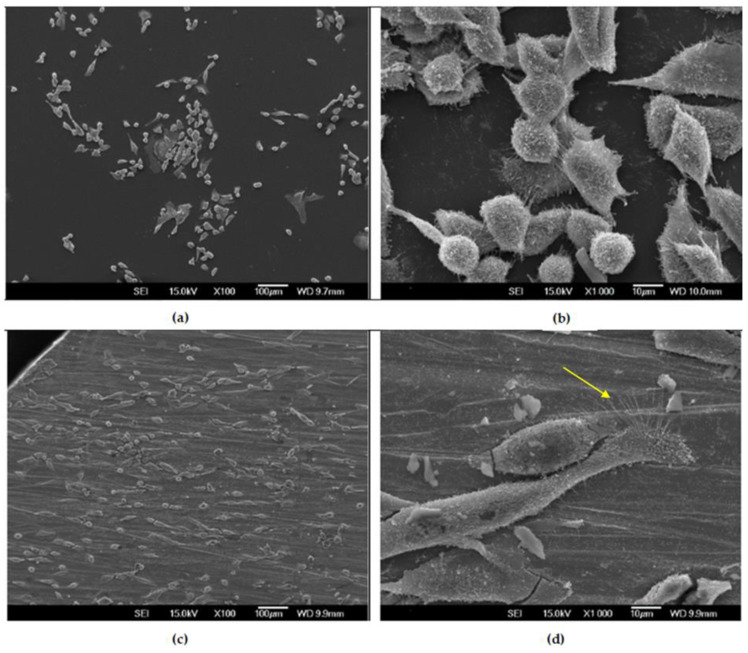
SEM micrographs of MG-63 cells after 24h cultured on discs: (**a**,**b**) Control disc; (**c**,**d**) Cu-Al-Ni disc. The yellow arrow indicates cells hair-like protrusions, i.e., filopodia.

**Figure 7 materials-16-02851-f007:**
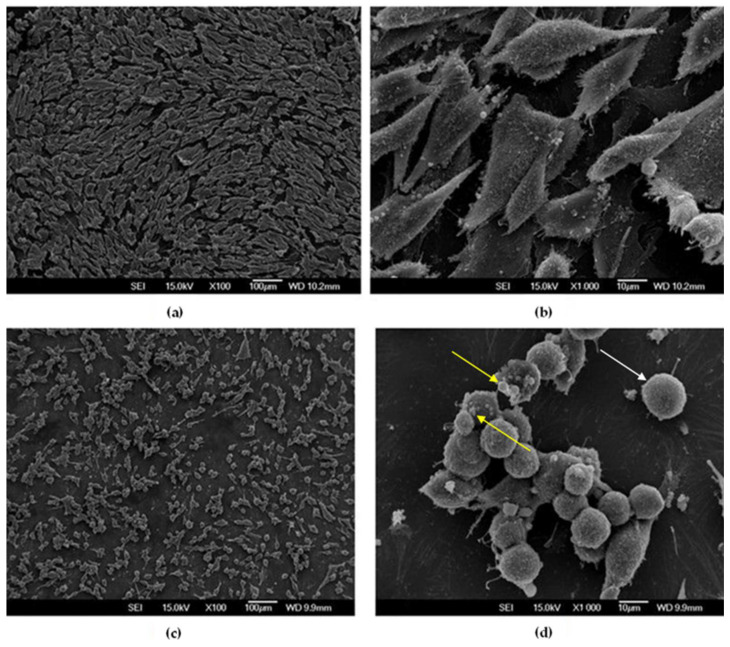
SEM micrographs of MG-63 cells after 7 days cultured on discs: (**a**,**b**) Control disc; (**c**,**d**) Cu-Al-Ni disc. The yellow arrows indicate apoptotic bodies. The white arrow indicates a round-shaped cell.

**Table 1 materials-16-02851-t001:** Chemical composition—XRF analysis.

	Cu	Al	Ni	Fe
Mean value (in wt.%)	Base	12	3.9	0.03

**Table 2 materials-16-02851-t002:** Chemical composition—EDX analyses.

Measuring Segment		1	2	3
Mean value	Cu	83.71 ± 0.30	83.81 ± 0.29	84.42 ± 0.67
in wt.%	Al	12.14 ± 0.32	12.20 ± 1.06	12.01 ± 0.84
±SD	Ni	4.15 ± 0.59	3.99 ± 0.91	3.58 ± 0.52

**Table 3 materials-16-02851-t003:** ASTM analysis results.

Grain Number G	Number of Grains per mm^2^	Mean Number of Intersections mm
3	64	0.113

**Table 4 materials-16-02851-t004:** Descriptive statistics for hardness and microhardness measurements.

	Mean ± SD	Min	Max	N
Hardness HV5	253 ± 12.43	229	266	8
Microhardness HV0.05	280 ± 19.50	260	299	3

## Data Availability

The data presented in this study are available on request from the corresponding author.

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
