# Peer review of "In Vitro Evaluation of the Potential Anticancer Properties of Cu-Based Shape Memory Alloys"

_materials, 2023, doi:10.3390/ma16072851_

Round 1
Author Response
Reviewer 1:
This manuscript is in good shape. I will recommend to publish after major revision.
Comment 1. There are so many grammatical mistakes.
Answer 1. Thank you for your comment. The manuscript has now been revised again by a native English speaker.
Comment 2. Figure (1,3) has to revise in terms of fonts and labelling.
Answer 2: The text on the figures has been corrected according to the instructions.
Comment 3. I would suggest to add phase transformation result.
Answer 3. This was not the subject of the investigation - there are no results.
Comment 4. I would recommend to work with more literature and modify result and discussion part because scientific reasoning and experimental validation behind various results are
missing
Answer 4. Thank you for the suggestion. We added more information and thus considerably expanded the Discussion section. Appropriate, new references were added as well:
- Čolić, M.; Rudolf, R.; Stamenković, D.; Anžel, I.; Vučević, D.; Jenko, M.; Lazić, V.; Lojen, G. Relationship between Microstructure, Cytotoxicity and Corrosion Properties of a Cu–Al–Ni Shape Memory Alloy. Acta Biomater. 2010, 6, 308–317, doi:10.1016/j.actbio.2009.06.027.
- Tomić, S.Z. Biocompatibility and immunomodulatory properties of nanomaterials based on gold nanoparticles and carbon nanotubes, and NITi-and CuAINi alloy-based advanced biomaterials. Ph.D. Thesis, University of Belgrade, Belgrade, Serbia, 2016. Available online: https://nardus.mpn.gov.rs/handle/123456789/6331
- Todorović, A.; Rudolf, R.; Romčević, N.; Đorđević, I.; Milošević, N.; Trifković, B.; Veselinović, V.; Čolić, M. BIOCOMPATIBILITY EVALUATION OF Cu-Al-Ni SHAPE MEMORY ALLOYS. Contemp. Mater. 2014, 5, 228–238, doi:10.7251/COMEN1402228T.
- A.C., K.; E., U.; Bruncko, M.; K., M.; Lojen, G.; H., S. Microstructure and Properties of Niti and Cualni Shape Memory Alloys. 2008, 14.
- Payandeh, Y.; Mirzakhani, B.; Bakhtiari, Z.; Hautcoeur, A. Precipitation and Martensitic Transformation in Polycrystalline CuAlNi Shape Memory Alloy – Effect of Short Heat Treatment. J. Alloys Compd. 2022, 891, 162046, doi:10.1016/j.jallcom.2021.162046

Reviewer 2 Report
1. EDS graph is to be included for confirmation of composition mentioned in the paper.
Author Response
Review 2
Comment:
- EDS graph is to be included for confirmation of composition mentioned in the paper.
Reply: EDS graph is now included in Figure 1.

Reviewer 3 Report
According to the reviewer, the article does not provide enough material for publication in a journal of the 2nd quartile. The reviewer recommends adding a study of 1-2 more chemical compositions or the effect of thermomechanical processing.
Article also requires major revision
The purpose of the work needs to be more clearly defined.
The planned chemical composition is not indicated - how much does it differ from the actual one?
Not substantiated, is this chemical composition the most optimal
It is necessary to describe in detail the technology of obtaining an alloy
The characteristic temperatures of martensitic transformations are not indicated.
X-ray lines need to be indexed
What are the reasons for such significant changes of XRD lines
The meaning of figure 4 is not clear
Author Response
Reviewer 3:
Comment 1. According to the reviewer, the article does not provide enough material for publication in a journal of the 2nd quartile. The reviewer recommends adding a study of 1-2 more chemical compositions or the effect of thermomechanical processing.
Answer 1. Thank you for the remark. We believe there is no need for additional experiments as the data from the literature are clear and coherent about the effect of chemical composition on the martensitic structure. According to the literature, this chemical composition is close to the optimal one, which is about 13 wt. % Al and 3 - 4.5 wt. %. If Ni content increases, the brittleness of the alloy increases, and the eutectoid point shifts to higher Al contents, which has a massive impact on biofunctional properties. (Lines 289-298).
In alloy Cu12Al4Ni (wt.%) phases β' (monoclinic), γ' (orthorhombic) are expected - both martensites thermally induced; the β' phase will be more dominant if the Al content is a little lower than this value, and the γ' phase if the Al content is a little higher than that.
Considering the chemical composition of our alloy, obtained by the EDX and XRF method, we expected those phases, and XRD analysis confirmed that. The weight percent of Al is 12%, and Ni is 3,9 % (SD > 1).
According to the literature data, the microstructure of β' martensite is characterized by the grain's needle "zig-zag" orientation. At the same time, the γ' phase contains coarse, the so-called "plate-like variants." Similar grains orientation are detected on optical micrographs in our study, but the plate-like variants are more dominant.
Article also requires major revision
Comment 2. The purpose of the work needs to be more clearly defined.
Answer 2. Thank you for your remark. We have tried to better define the purpose of the study and also to comment the rationale of the study in a more detailed way, in the Discussion section.
This study aimed to examine the surface microstructure, structure, hardness and cytotoxic effects of a Cu-Al-Ni alloy on human osteosarcoma cells (so-called investigation of the biofunctional characteristics-Lines 72-79).
The Cu-Al-Ni alloy can be potentially used for the preparation of medical devices. As different studies point to a better control of chemical composition in comparison to NiTi alloy, better ageing resistance and also a better stability of the two-way shape memory effect.
As the toxicity of metal biomaterials is proportional to released metal ions, depending on the chemical composition and fabrication, Cu-Al-Ni alloys can show different cytocompatibility/cytotoxiyity. Prior to this investigation, we performed a biocompatibility study, and the results indicated that CuAlNi alloy did not elicit cytotoxicity against two normal human cell lines (fibroblasts and dental pulp cells) measured with standard cytotoxicity tests. Based on contemporary scientific interest in Cu as an antitumor agent, and given the application of Cu-based shape memory alloys in biomedical engineering, some chemotherapy treatments are based on biological MEMS for drug delivery. The working principle of these bio-actuators and sensors is based on the two-way shape memory effect (TWSME). These devices develop micro-mechanical movements triggered by external stimuli changes (temperature, electromagnetic or fluid-driven).The uniqueness of these stimuli-response devices is their ability to sense and actuate thanks to the materials' phase change. As a consequence, these systems can control the dosage of the released drug in a target tissue. A novel therapeutic treatment is based on the fact that implantable MEMS have prolonged release of drugs into the body from micro-chamber. However, the literature is poor with data about the effect of these materials on cancer cells itself (this new information has been added in Lines 270-279).
To the best of our knowledge, the topic of anticancer properties of CuAlNi alloy has not previously been examined.
Comment 3. The planned chemical composition is not indicated - how much does it differ from the actual one?
Answer 3.The target alloy composition was about 13 wt.% Al and 3 - 4.5 wt.%, and Cu base. The obtained results from the investigated Cu-Al-Ni discs were 12 wt% Al, 3,9wt % Ni, and Cu base.
Comment 4. Not substantiated, is this chemical composition the most optimal
Answer 4. Yes, it is the most optimal.
Comment 5. It is necessary to describe in detail the technology of obtaining an alloy.
Answer 5.
The manufacturing technology is explained by reference no. 15:
- Lojen, G.; Stambolić, A.; Šetina Batič, B.; Rudolf, R. Experimental Continuous Casting of Nitinol. Metals (Basel).2020, 10, 505, doi:10.3390/met10040505.
Added text: The Cu-Al-Ni alloy was prepared by vacuum induction melting of pure Cu, Al, Ni components immediately before casting. Experimental casting about 1300°C was performed with a laboratory scale vertical continuous casting (VCC) device, Technica Guss, which was connected to the 60 kW medium-frequency (4 kHz) vacuum induction melting (VIM) furnace Leybold Hereaus. The charge was approximately 15 kg. The induction power was 10 kW for the first 10 min; for the next 10 min 20 kW; and in the final 5 min, 30 kW. Melting was carried out in a vacuum. Just before the start of casting, the chamber was filled with argon (purity 5.0). To minimise the risk of sticking and fracture of the rod in the mould, relatively low casting rates were selected, and the pulling sequence was programmed with the lowest possible acceleration.
Comment 6. The characteristic temperatures of martensitic transformations are not indicated.
Answer 6: This was not the subject of this study.
Comment 7. X-ray lines need to be indexed–
Answer 7: Improved.
What are the reasons for such significant changes of XRD lines
Answer 7: Hypothesis: The manufacturing process i.e. melting and casting conditions were not at equilibrium for formation of stable phases in order to obtain a stoichiometric chemical composition.
Comment 8. The meaning of figure 4 is not clear
Answer 8. Thank you for the suggestion, the Figure 4 has been deleted.

Round 2
Reviewer 3 Report
The authors have significantly improved the article and it can be published.
The reviewer advises to indicate (since the authors do not want to present the results of the DSC) the approximate temperatures of martensitic transformations in the introduction. Is the alloy in a superelastic state at body temperature
Author Response
Reviewer 3:
The reviewer advises to indicate (since the authors do not want to present the results of the DSC) the approximate temperatures of martensitic transformations in the introduction. Is the alloy in a superelastic state at body temperature
Explanation:
Superelastic effect is connected at constant temperature as a function of the stress. It is the property or ability of a material to recover a very high elongation ~ 6-10 % to its original position or shape spontaneously upon unloading. Superelasticity usually occurs in the temperature range between Af < T < Md. Moreover, superelasticity is usually induced when a material is deformed at a temperature higher than its Af temperature, where an external stress causes a stress transformation of the parent austenite phase into a transformed martensite phase. Given that Af for this CuAlNi alloy is above 200°C [1], this alloy is not in a superelastic state at body temperature.
Reference:
- López-Ferreño, J.F. Gómez-Cortés, T. Breczewski, I. Ruiz-Larrea, M.L. Nó, J.M. San Juan,
High-temperature shape memory alloys based on the Cu-Al-Ni system: design and thermomechanical characterization, Journal of Materials Research and Technology, Volume 9, Issue 5, 2020, Pages 9972-9984, ISSN 2238-7854, https://doi.org/10.1016/j.jmrt.2020.07.002.
